# ST-VLO: Unified Spatio-Temporal Correlation for Visual-LiDAR Odometry with Temporal Drift Compensation

## Abstract

We propose an effective and efficient visual-LiDAR odometry framework named ST-VLO, which establishes the unified spatio-temporal correlation with Mamba models and addresses the long-standing cumulative drift problem with temporal compensation for the localization in 4D dynamic environments. Specifically, ST-VLO includes a novel unified spatial-temporal correlation module established on Mamba to fuse heterogeneous visual and LiDAR information across multi-frame video clips, overcoming the insufficient temporal information exploration in previous pairwise odometry methods. Furthermore, a Temporal Drift Compensation module is designed to minimize cumulative drifts by iteratively learning correction residuals from multiple history frames. To strengthen the spatial feature representation on salient features, we also propose a Keypoint-Aware Auxiliary Loss with a winner-takes-all strategy. ST-VLO achieves state-of-the-art performance on two commonly-used autonomous driving datasets, surpassing previous methods with a 19% $t_{rel}$ and 22% $r_{rel}$ reduction on KITTI, and a 18% ATE and 16% RPE reduction on Argoverse.

## 1 Introduction

The odometry task aims to estimate relative pose transformations between consecutive frames in autonomous systems, which has various downstream applications, such as self-driving vehicles (Wang et al., 2021a; Hu et al., 2023; Jiang et al., 2023), SLAM systems (Lipson et al., 2024b; Teed et al., 2024; Deng et al., 2024; Teed & Deng, 2021; Campos et al., 2021; Yuan et al., 2023; 2024b; Zheng et al., 2022; 2024), and robot navigation (Liu et al., 2024b;c).

Recently, multi-modal approaches (Wang et al., 2021b; Graeter et al., 2018; Zhuoins et al., 2023; Liu et al., 2024a) have gained significant attention for improving odometry estimation, which combine visual (Kendall et al., 2015; Wang et al., 2017; Shan et al., 2024; Deng et al., 2023b; Wang et al., 2024b) and LiDAR (Chen et al., 2023; Yuan et al., 2022; Wang et al., 2022b;a; Zhang et al., 2024b; Deng et al., 2023a; Wang et al., 2019) data to address issues like structure misalignment (Liu et al., 2024a), single sensor degradation (Deng et al., 2023b), and poor robustness in dynamic outdoor environments (Wang et al., 2021b). However, most existing odometry frameworks primarily rely on pairwise frame inputs (Liu et al., 2024a; 2023a; Li et al., 2022; Xu et al., 2022; Li et al., 2019), neglecting essential temporal information within multi-frame sequences.

In this paper, we present **ST-VLO**, which efficiently and effectively fuses multi-modal data from both images and LiDAR. Crucially, rather than limiting the estimation process to isolated source-target frame pairs while discarding previously observed frames, we propose a novel method that not only estimates relative motion from consecutive frames but also utilizes motion priors from history frames. We introduce a differentiable mechanism that enables end-to-end learning, allowing the model to retrieve and incorporate long-term temporal cues from the entire LiDAR sequences.

ST-VLO seamlessly integrates heterogeneous multi-modal fusion and temporal modeling through a unified *MMG (MaxPooling, Mamba (Gu & Dao, 2024), gMLP (Liu et al., 2021a))* architecture, inspired by sparse deformable query fusion (Zhu et al., 2021). As shown in Fig. 1, ST-VLO captures spatial and temporal dependencies by fusing heterogeneous inputs adaptively, creating a cohesive representation that enables precise pose estimation over extended sequences. Unlike existing pairwise methods, ST-VLO leverages multi-frame video clips, aggregating history poses and features

Figure 1: **Comparison with previous odometry methods.** Previous methods typically encode spatial and temporal features separately, relying on pairwise correlations between consecutive frames. In contrast, we propose a unified module to jointly extract spatial and temporal features, and a differentiable temporal compensation to mitigate accumulative drifts.

with *MMG* to model long-term dependencies effectively. To further reduce accumulative drifts, ST-VLO employs a novel *Temporal Drift Compensation* technique, which minimizes drifts iteratively by directly learning the residual error reduction over certain frame intervals. Additionally, to achieve robust spatial understanding and be less susceptible to noise in sequential data, ST-VLO incorporates a *Keypoint-Aware Auxiliary Loss*, which refines high-salient feature regions by selectively optimizing the top-k queries with the smallest error relative to the ground truth pose, using a winner-takes-all loss strategy (Makansi et al., 2019). Based on these proposed components, ST-VLO serves as a systematical and scalable solution for low-drift odometry across dynamic environments.

Overall, our **key contributions** are as follows:

- We propose a novel visual-LiDAR odometry network named ST-VLO. A modality-agnostic *MMG* architecture captures unified spatial information in visual and LiDAR data and long-term temporal dependencies across multiple frames for accurate pose estimation.

- The *Temporal Drift Compensation* mechanism predicts cumulative pose errors over a sequence of frames, effectively compensating for the pairwise motion estimation to address the drift problem of multi-frame temporal inputs.

- We design a *Keypoint-Aware Auxiliary Loss* to selectively optimize high-salient features focusing on regions associated with static objects for stable reference, refining 3D spatial representations for accurate pose estimation.

- Extensive experiments on the KITTI odometry (Geiger et al., 2012) and Argoverse dataset (Chang et al., 2019b) show that our method surpasses all recent deep learning-based LiDAR, visual, and visual-LiDAR-fusion odometry approaches across most sequences.

## 2 RELATED WORK

### 2.1 VISUAL-LIDAR ODOMETRY

Visual-LiDAR odometry combines the strengths of visual and LiDAR sensors, leveraging 2D texture (Huang et al., 2025; Bie et al., 2025) and 3D geometric (Lai et al., 2025; Li et al., 2025; Zhao et al., 2025) information to improve pose estimation. Previous approaches are broadly classified into loosely and tightly integrated methods. Loosely integrated methods (Zhang et al., 2017; Graeter et al., 2018; Shin et al., 2020; Huang et al., 2020a) use LiDAR data to enhance depth estimation and visual data for pose tracking. However, these approaches rely on interpolation between 3D points and 2D pixels, which introduces potential errors due to the inaccurate point-to-pixel correspondences. Tightly integrated methods (Zhang & Singh, 2015; An et al., 2022; Shubodh et al., 2024) aim for a seamless fusion of visual and LiDAR data to enhance consistency. For the learning-based methods, MVL-SLAM (An et al., 2022) fuses RGB images and LiDAR using RCNNs. LIP-Loc (Shubodh et al., 2024) applies contrastive learning for cross-modal localization. Nevertheless, they struggle with the structural differences between point clouds and images. Recent works (Lai et al., 2022; Liu et al., 2024a) introduce descriptor fusion or clustering techniques to further improve the structural alignment. However, these approaches often suffer from increasing computational costs due to the additional feature processing, limiting their real-time applicability in practical scenarios. Moreover, previous methods are commonly constrained by pairwise inputs, which, however, lack multi-frame temporal information. To our knowledge, our work is the first visual-LiDAR

Figure 2: **Overview of the ST-VLO framework.** The unified MMG (Maxpooling, Mamba, gMLP) module fuses visual and LiDAR features with Deformable Mamba, while Temporal Mamba leverages memory banks for long-term modeling. The predicted pose is iteratively refined and further corrected through Temporal Drift Compensation.

odometry method designing Deformable Mamba and Temporal Mamba for efficient query-based fusion across multiple frames, thereby enabling a unified multi-modal representation.

## 2.2 MEMORY MECHANISMS FOR AUTONOMOUS DRIVING

Memory-based temporal modeling is crucial for robust driving under occlusions and sensor dropouts (Han et al., 2023; Yang et al., 2023; Yuan et al., 2024a). Feature-level approaches like BEVFormerv2 (Yang et al., 2023) and VideoBEV (Han et al., 2023) propagate BEV queries across frames, while vectorized models such as Sparse4Dv2/3 (Lin et al., 2023a;b) adopt recurrent or denoising strategies for object queries. These designs highlight the importance of temporal consistency and have also inspired HD mapping frameworks (Yuan et al., 2024a). Our method builds upon these insights by maintaining both implicit feature memory and explicit pose memory to handle long-term temporal dependencies.

## 2.3 TEMPORAL MODELING WITH MAMBA

Sequential models for forecasting and planning often rely on RNNs (Salzmann et al., 2020; Varadarajan et al., 2022) or Transformers (Vaswani et al., 2017; Ngiam et al., 2022), but state space models (SSMs) (Gu et al., 2021a;b; Ke et al., 2025; Liu et al., 2025a) are emerging as efficient alternatives. Mamba (Gu & Dao, 2024) introduces selective SSMs with hardware-friendly pipelines, while VMamba (Liu et al., 2024d) and Mamba-ND (Li et al., 2024) extend to images and multi-dimensional data. Recent works further integrate SSMs into diffusion models for efficient generation (Yan et al., 2023). Inspired by these, we combine Mamba-based implicit feature memory with explicit pose memory to achieve more robust temporal modeling.

## 3 METHOD

The architecture of ST-VLO is shown in Fig. 2. ST-VLO begins with the *Unified Spatio-Temporal Correlation* (Sec. 3.1) with deformable spatial Mamba and temporal Mamba fusion, which integrates visual and LiDAR features across frames into a unified representation. This unified representation is utilized to predict the pose estimates. In addition, the predicted pose is compensated with long-term temporal information using *Temporal Drift Compensation* (Sec. 3.2) to mitigate error accumulation, refining the final pose. The loss functions including our designed *Keypoint-Aware Auxiliary Loss* with a winner-takes-all strategy is specified in Sec. 3.3.

Before we delve into the detailed model design, we follow Wang et al. (2022a); Liu et al. (2023a; 2024a) for point encoding and (Huang et al., 2020b) for image encoding to extract hierarchical features for both modalities. 3D points are first projected onto a cylindrical surface (Wang et al., 2022a) to organize the originally irregular points, where the original 3D coordinates are further filled in corresponding projected 2D positions to preserve raw point geometry, converting the point cloud into pseudo-images of size $H_P \times W_P \times 3$. Then these pseudo-images are passed to a point encoder (Wang et al., 2022a), producing multi-scale point features represented as $\mathbf{F}_P \in \mathbb{R}^{H_P \times W_P \times D}$, where $D$ is

the feature dimension. The 2D input images $I \in \mathbb{R}^{H \times W \times 3}$ are processed through a convolutional feature pyramid network (Liu et al., 2024a), generating multi-level image features represented as $\mathbf{F}_I \in \mathbb{R}^{H \times W \times C}$, where $H$, $W$, and $C$ correspond to the height, width, and channels, respectively.

## 3.1 UNIFIED SPATIO-TEMPORAL CORRELATION

We design a Unified Spatio-Temporal Correlation block consisting of MaxPooling, Mamba (Gu & Dao, 2024), and gMLP (Liu et al., 2021a) (*MMG*) to extract spatial and temporal representations of point and image features. It harmonizes diverse input representations across different modalities and temporal dimensions: The gMLP encodes sequential information into a unified feature space, providing a foundation for learning from varied data sources. Mamba then establishes temporal interactions within the sequences, capturing long-term dependencies across frames. We extend the Mamba structure into a *Deformable Mamba* for spatial modeling and a *Temporal Mamba* for temporal modeling, in order to achieve effective and efficient modality-agnostic modeling by harnessing diverse input representations across modalities and temporal dimensions, which are detailed below. Finally, MaxPooling condenses the sequence into a single, unified representation, preserving essential information in a compact form.

**Deformable Mamba.** We propose a deformable Mamba-based feature fusion module by extending the concept of deformable attention proposed in Deformable DETR (Zhu et al., 2021). As shown in Fig. 2, LiDAR features $\mathbf{F}_P$, which encode 3D spatial information, serve as queries to efficiently combine LiDAR and camera data. Specifically, we first project the LiDAR point onto the image plane using the camera intrinsic and extrinsic parameters, obtaining the reference point. Then, we sample the visual features $\mathbf{F}_{\text{sample}}$ using a bilinear interpolation from the image features $\mathbf{F}_I$ around the reference point with adaptive offsets. Consequently, the sampled visual features $\mathbf{F}_{\text{sample}}$ are fused with the LiDAR features $\mathbf{F}_P$ using *MMG* as:

$$\mathbf{F}_{\text{fused}} = \text{MaxPool}(\text{Mamba}(G_f(\mathbf{F}_{\text{sample}} \oplus \mathbf{F}_P))), \tag{1}$$

where $\oplus$ denotes concatenation, $G_f$ is the gMLP layer, and Mamba refers to the standard Mamba block in (Gu & Dao, 2024). This multi-modal fusion enriches the LiDAR feature space with visual context while maintaining efficiency by focusing on adaptive local receptive fields.

Finally, the fused multi-modal features $\mathbf{F}_{\text{fused}}$ from consecutive frame pairs are correlated by the cost volume module (Wang et al., 2021a; 2022a), producing a cross-frame motion feature $\mathbf{E}_{\text{ego}} \in \mathbb{R}^{N \times D}$, where $N$ is the number of downsampled points in the coarsest layer.

**Temporal Mamba.** To effectively model the temporal information, Mamba-based Memory Feature Bank (MFB) and Memory Pose Bank (MPB) are leveraged to store historical feature representations and pose information.

In the MFB, we store historical ego-motion features from the Deformable Mamba above, namely $[\mathbf{E}_{\text{ego}, t-T_h+1}, \ldots, \mathbf{E}_{\text{ego}, t-1}]$, which integrate historical semantic and geometric contexts. An *MMG* with another gMLP $G_e$ enables effective temporal interactions by first including the current frame's ego-motion feature $\mathbf{E}_{\text{ego},t}$ into the MFB, forming the history list $\mathcal{F} = [\mathbf{E}_{\text{ego}, t-T_h+1}, \ldots, \mathbf{E}_{\text{ego}, t}] \in \mathbb{R}^{N \times T_h \times D}$, and then computing the updated ego-motion feature:

$$\hat{\mathbf{E}}_{\text{ego}} = \text{MaxPool}\big(\text{Mamba}\big(G_e(\mathcal{F})\big)\big). \tag{2}$$

In the MPB, we store historical quaternions $\mathcal{Q} = [\mathbf{q}_{t-T_h+1}, \ldots, \mathbf{q}_{t-1}]$ and translations $\mathcal{P} = [\mathbf{p}_{t-T_h+1}, \ldots, \mathbf{p}_{t-1}]$. Another pair of *MMG*s, with $G_q$ and $G_p$, respectively, encodes these historical pose states, yielding temporal embeddings $\mathbf{Q}_{\text{enc}}$ and $\mathbf{P}_{\text{enc}}$:

$$\mathbf{Q}_{\text{enc}} = \text{MaxPool}\big(\text{Mamba}\big(G_q(\mathcal{Q})\big)\big), \quad \mathbf{P}_{\text{enc}} = \text{MaxPool}\big(\text{Mamba}\big(G_p(\mathcal{P})\big)\big). \tag{3}$$

Both MFB and MPB are initialized with zeros and iteratively updated as new frames are sequentially observed, as shown in Fig. 2.

Finally, we construct a unified scene representation $\{\mathbf{E}_{\text{ego}}, \hat{\mathbf{E}}_{\text{ego}}, \mathbf{Q}_{\text{enc}}, \mathbf{P}_{\text{enc}}\}$ by unifying the multi-modal and long-term temporal information across these *MMG*s. The initial quaternion $\mathbf{q}^{(1)} = \frac{\Phi_q(\hat{\mathbf{E}}_{\text{ego}} \oplus \mathbf{Q}_{\text{enc}})}{\|\Phi_q(\hat{\mathbf{E}}_{\text{ego}} \oplus \mathbf{Q}_{\text{enc}})\|}$ is calculated as the normalized output of the MLP $\Phi_{\mathbf{q}}$, while the initial translation $\mathbf{p}^{(1)} = \Phi_p(\hat{\mathbf{E}}_{\text{ego}} \oplus \mathbf{P}_{\text{enc}})$ is predicted using the MLP $\Phi_p$. We then refine these through an

iterative refinement module in multiple upper layers (Wang et al., 2021a), obtaining the refined pose $\mathbf{q}^{(2)}, \mathbf{p}^{(2)}$ at each time step.

## 3.2 TEMPORAL DRIFT COMPENSATION

After obtaining finer frame-to-frame poses $\mathbf{p}^{(2)}$ and $\mathbf{q}^{(2)}$, ST-VLO also incorporates an optimization technique here to reduce the cumulative errors to achieve low-drift and more precise long-range odometry estimation as shown in Fig. 3. Specifically, we accumulate multiple pose estimates through the cumulative multiplication of poses from the nearest historical $T_g$ frames: $(\mathbf{q}_t^{\text{cumul}}, \mathbf{p}_t^{\text{cumul}}) = (\mathbf{q}_t^{(2)}, \mathbf{p}_t^{(2)}) \circ (\mathbf{q}_{t-1}^{(2)}, \mathbf{p}_{t-1}^{(2)}) \circ \cdots \circ (\mathbf{q}_{t-T_g+1}^{(2)}, \mathbf{p}_{t-T_g+1}^{(2)})$, where the operation $\circ$ is calculated as $\mathbf{q}_{a \circ b} = \mathbf{q}_a * \mathbf{q}_b$ and $[0, \mathbf{p}_{a \circ b}] = \mathbf{q}_a [0, \mathbf{p}_b] \mathbf{q}_a^{-1} + [0, \mathbf{p}_a]$, with $*$ denoting the quaternion product.

Given the source point cloud data ($\text{PC}_{T_t - T_g} \in \mathbb{R}^{N \times 3}$) back to the time step $T_t - T_g$, we then use the accumulative pose estimates above to warp it to the current frame $T_t$ as follows:

$$[0, \widehat{\text{PC}}_{T_t}] = \mathbf{q}_t^{\text{cumul}} [0, \text{PC}_{T_t - T_g}] (\mathbf{q}_t^{\text{cumul}})^{-1} + [0, \mathbf{p}_t^{\text{cumul}}]. \tag{4}$$

Subsequently, at the current step, the residual pose error $(\Delta \mathbf{q}^{(2)}, \Delta \mathbf{p}^{(2)})$ between the warped source point cloud $\widehat{\text{PC}}_{T_t}$ and the target point cloud $PC_{T_{t+1}}$ is calculated by the Pyramid, Warping, Cost volume (PWC) structure (Wang et al., 2021a).

Finally, as shown in the lower part of Fig. 2, the residual pose error is incorporated into the current pose at $T_t$ to iteratively refine the final pose as follows (the time step is omitted below for simplicity):

$$\mathbf{q}^{(3)} = \Delta \mathbf{q}^{(2)} * \mathbf{q}^{(2)}, \quad [0, \mathbf{p}^{(3)}] = \Delta \mathbf{q}^{(2)} [0, \mathbf{p}^{(2)}] (\Delta \mathbf{q}^{(2)})^{-1} + [0, \Delta \mathbf{p}^{(2)}]. \tag{5}$$

Compared to pairwise estimated pose errors, point cloud warping allows backtracking to much earlier observed point cloud data. This enables the compensation mechanism to amplify pose errors across a sequence of accumulated frames, which penalizes cumulative drift and in turn enhances alignment accuracy over multiple frames. During training, to fully optimize model parameters, we compute the compensation loss for cumulative pose error whenever the history of pose frames exceeds $T_g$. During inference, this is applied to cumulative poses every $T_g$ frames to maintain efficiency. This compensation loss is embedded into the regression loss for pose estimation, as specified in equation 6.

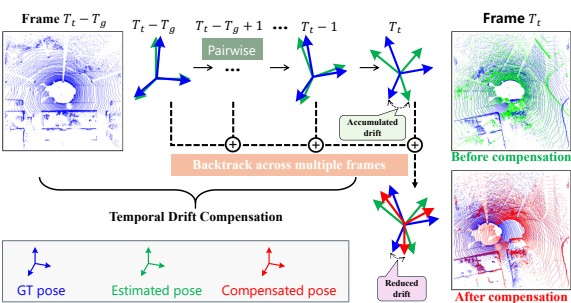

Figure 3: **Illustration of the Temporal Drift Compensation.**

## 3.3 LOSS FUNCTION

The overall loss function mainly consists of three parts:

**Regression Loss.** We predict poses at three stages: initial $(\mathbf{q}^{(1)}, \mathbf{p}^{(1)})$, refined $(\mathbf{q}^{(2)}, \mathbf{p}^{(2)})$, and drift-compensated $(\mathbf{q}^{(3)}, \mathbf{p}^{(3)})$. For each stage, the loss follows Wang et al. (2022a); Liu et al. (2024a):

$$\mathcal{L} = \|\hat{\mathbf{p}} - \mathbf{p}\| \exp(-k_t) + k_t + \|\hat{\mathbf{q}} - \mathbf{q}\|_2 \exp(-k_q) + k_q, \tag{6}$$

where $k_t, k_q$ are learnable scalars for translation ($L^1$) and rotation ($L^2$); $\hat{\mathbf{p}}, \hat{\mathbf{q}}$ denote ground truth and $\mathbf{p}, \mathbf{q}$ predictions. For the residual pose output from the *Temporal Drift Compensation* module, we narrow differences between ground truth cumulative pose $(\hat{\mathbf{q}}^{\text{cumul}}, \hat{\mathbf{p}}^{\text{cumul}})$ and predicted cumulative pose $(\mathbf{q}^{\text{cumul}}, \mathbf{p}^{\text{cumul}})$. The total regression loss is a weighted sum over stages, i.e., $\mathcal{L}^{\text{reg}} = \alpha^1 \mathcal{L}^{(1)} + \alpha^2 \mathcal{L}^{(2)} + \alpha^3 \mathcal{L}^{(3)}$. where $(\mathcal{L}^{(1)}, \mathcal{L}^{(2)}, \mathcal{L}^{(3)})$ are computed from $(\mathbf{q}^{(1)}, \mathbf{p}^{(1)})$, $(\mathbf{q}^{(2)}, \mathbf{p}^{(2)})$, and $(\mathbf{q}^{(3)}, \mathbf{p}^{(3)})$, respectively.

**Keypoint-Aware Auxiliary Loss.** Learning discriminative features with strong robustness to noise is crucial for our odometry accuracy, as ST-VLO relies on feature-based pose regression. To

Table 1: **Comparison with different odometry methods on the KITTI odometry (Geiger et al., 2013).** $t_{rel}$ and $r_{rel}$ represent the average sequence translational RMSE (%) and sequence rotational RMSE (°/100m) respectively in the length of 100, 200, ..., 800m. The best results are **bold**, and the second best results are underlined. * represents the models trained on the 00-08 sequences.

| Method | 00 | | 01 | | 02 | | 03 | | 04 | | 05 | | 06 | | 07 | | 08 | | 09 | | 10 | | mean (07-10) | |
|---|---|---|---|---|---|---|---|---|---|---|---|---|---|---|---|---|---|---|---|---|---|---|---|---|
| | $t_{rel}$ | $r_{rel}$ | $t_{rel}$ | $r_{rel}$ | $t_{rel}$ | $r_{rel}$ | $t_{rel}$ | $r_{rel}$ | $t_{rel}$ | $r_{rel}$ | $t_{rel}$ | $r_{rel}$ | $t_{rel}$ | $r_{rel}$ | $t_{rel}$ | $r_{rel}$ | $t_{rel}$ | $r_{rel}$ | $t_{rel}$ | $r_{rel}$ | $t_{rel}$ | $r_{rel}$ | $t_{rel}$ | $r_{rel}$ |
| Visual Odometry Methods | | | | | | | | | | | | | | | | | | | | | | | | |
| SfMLearner* (Zhou et al., 2017) | 21.32 | 6.19 | 22.41 | 2.79 | 24.10 | 4.18 | 12.56 | 4.52 | 4.32 | 3.28 | 12.59 | 4.66 | 15.55 | 5.58 | 12.61 | 6.31 | 10.66 | 3.75 | 11.32 | 4.07 | 15.25 | 4.06 | 12.46 | 4.55 |
| DFVO* (Zhan et al., 2021) | 2.01 | 0.79 | 61.17 | 18.96 | 2.46 | 0.79 | 3.27 | 0.89 | 0.79 | 0.56 | 1.50 | 0.74 | 1.95 | 0.76 | 2.28 | 1.16 | 2.11 | 0.74 | 3.21 | 0.59 | 2.89 | 0.97 | 2.62 | 0.87 |
| Cho et al.* (Cho & Kim, 2023) | 1.77 | 0.79 | 64.38 | 16.87 | 2.62 | 0.74 | 3.06 | 0.89 | 0.65 | 0.55 | 1.31 | 0.74 | 1.60 | 0.56 | 1.06 | 0.67 | 2.28 | 0.76 | 2.66 | 0.53 | 2.95 | 0.95 | 2.24 | 0.73 |
| LiDAR Odometry Methods | | | | | | | | | | | | | | | | | | | | | | | | |
| LO-Net (Li et al., 2019) | 1.47 | 0.72 | 1.36 | 0.47 | 1.52 | 0.71 | 1.03 | 0.66 | 0.51 | 0.65 | 1.04 | 0.69 | 0.71 | 0.50 | 1.70 | 0.89 | 2.12 | 0.77 | 1.37 | 0.58 | 1.80 | 0.93 | 1.75 | 0.79 |
| PWCLO (Wang et al., 2021a) | 0.89 | 0.43 | 1.11 | 0.42 | 1.87 | 0.76 | 1.42 | 0.92 | 1.15 | 0.94 | 1.34 | 0.71 | 0.60 | 0.38 | 1.16 | 1.00 | 1.68 | 0.72 | 0.88 | 0.46 | 2.14 | 0.71 | 1.47 | 0.72 |
| DELO (Ali et al., 2023) | 1.43 | 0.81 | 2.19 | 0.57 | 1.48 | 0.52 | 1.38 | 1.10 | 2.45 | 1.70 | 1.27 | 0.64 | 0.83 | 0.35 | 0.58 | 0.41 | 1.36 | 0.64 | 1.23 | 0.57 | 1.53 | 0.90 | 1.18 | 0.63 |
| TransLO (Liu et al., 2023a) | 0.85 | 0.38 | 1.16 | 0.45 | 0.88 | 0.34 | 1.00 | 0.71 | 0.34 | 0.18 | 0.63 | 0.41 | 0.73 | 0.31 | 0.55 | 0.43 | 1.29 | 0.50 | 0.95 | 0.46 | 1.18 | 0.61 | 0.99 | 0.50 |
| EfficientLO (Wang et al., 2022a) | 0.80 | 0.37 | 0.91 | 0.40 | 0.94 | 0.32 | 0.51 | 0.43 | 0.38 | 0.30 | 0.57 | 0.33 | 0.36 | 0.23 | 0.37 | 0.26 | 1.22 | 0.48 | 0.87 | 0.38 | 0.91 | 0.50 | 0.86 | 0.41 |
| DSLO (Zhang et al., 2024a) | 0.78 | 0.40 | **0.66** | 0.23 | 0.77 | 0.34 | 0.67 | 0.37 | 0.31 | 0.47 | 0.50 | 0.30 | 0.57 | 0.38 | 0.58 | 0.41 | 1.16 | 0.51 | 0.72 | 0.33 | 1.29 | 0.49 | 0.94 | 0.44 |
| Zhou et al. (Zhou et al., 2025) | — | — | — | — | — | — | — | — | — | — | — | — | — | — | 0.50 | 0.40 | 1.30 | 0.60 | 1.16 | 0.61 | 1.21 | 0.68 | 1.04 | 0.57 |
| LAGLO* (Tang et al., 2025) | 1.21 | **0.27** | 1.92 | 0.37 | 1.99 | 0.36 | 1.16 | 0.45 | 0.63 | 0.38 | 0.93 | 0.21 | 1.26 | 0.38 | 1.02 | 0.30 | 1.48 | 0.29 | 2.29 | 0.55 | 2.00 | 0.61 | 2.15 | 0.56 |
| Multi-modal Odometry Methods | | | | | | | | | | | | | | | | | | | | | | | | |
| An et al.* (An et al., 2022) | 2.53 | 0.79 | 3.76 | 0.80 | 3.95 | 1.05 | 2.75 | 1.39 | 1.81 | 1.48 | 3.49 | 0.79 | 1.84 | 0.83 | 3.27 | 1.51 | 2.75 | 1.61 | 3.70 | 1.83 | 4.65 | 0.51 | 3.59 | 1.37 |
| H-VLO* (Aydemir et al., 2022) | 1.75 | 0.62 | 4.32 | 0.46 | 2.32 | 0.60 | 2.52 | 0.47 | 0.73 | 0.36 | 0.85 | 0.35 | 0.75 | 0.30 | 0.79 | 0.48 | 1.35 | 0.38 | 1.89 | 0.34 | 1.39 | 0.52 | 1.36 | 0.43 |
| DVLO (Liu et al., 2024a) | 0.80 | 0.35 | 0.85 | 0.33 | 0.81 | 0.29 | 0.59 | 0.36 | 0.26 | 0.13 | 0.41 | 0.23 | 0.33 | **0.17** | 0.46 | 0.33 | 1.09 | 0.44 | 0.85 | 0.36 | 0.88 | 0.46 | 0.82 | 0.41 |
| DVLO4D (Liu et al., 2025b) | 0.68 | 0.33 | 0.77 | 0.23 | 0.76 | 0.31 | 0.49 | 0.33 | 0.22 | 0.13 | 0.39 | 0.21 | 0.32 | 0.21 | 0.43 | 0.32 | 0.95 | 0.36 | 0.77 | 0.33 | 0.76 | 0.46 | 0.73 | 0.37 |
| **ST-VLO (Ours)** | **0.59** | **0.27** | 0.73 | **0.21** | **0.65** | **0.23** | **0.47** | **0.31** | **0.21** | **0.12** | **0.33** | **0.15** | **0.31** | **0.19** | **0.26** | **0.23** | **0.79** | **0.28** | **0.63** | **0.28** | **0.65** | **0.39** | **0.59** | **0.29** |

strengthen feature learning, we introduce an auxiliary loss focusing on critical areas in the feature space. Specifically, we leverage the cost volume $E = \{e_i \mid e_i \in \mathbb{R}^c\}_{i=1}^N$ to predict pose $(\mathbf{q}^{key} \in \mathbb{R}^{N \times 4}, \mathbf{p}^{key} \in \mathbb{R}^{N \times 3})$ for each query in the feature map. We apply a winner-takes-all loss (Makansi et al., 2019) to optimize only the top-k queries $(\mathbf{q}^{key} \in \mathbb{R}^{K \times 4}, \mathbf{p}^{key} \in \mathbb{R}^{K \times 3})$ with the smallest error relative to the ground truth pose. As demonstrated in Fig. 6, this selective loss function helps the model focus on static regions, with the auxiliary loss defined as $\mathcal{L}^{aux} = \frac{1}{K} \sum_{k=1}^{K} \mathcal{L}^k$. where $\mathcal{L}^k$ indicates the regression losses between the poses of the top selected $k$ queries with the ground truth pose as calculated in the equation 6.

**Collective Average Loss.** We employ the Collective Average Loss (CAL) inspired by MOTR (Zeng et al., 2022), which aggregates losses across multiple frames. Specifically, the total loss of ST-VLO $\mathcal{L}_{total}$ is computed as the average loss over frames within each sub-clip $T_s$, i.e., $\mathcal{L}_{total} = \frac{1}{T_s} \sum_{t=1}^{T_s} \mathcal{L}_t$, where $\mathcal{L}_t = \mathcal{L}_t^{reg} + \alpha^4 \mathcal{L}_t^{aux}$.

# 4 EXPERIMENT

## 4.1 DATASETS AND METRICS

**KITTI Odometry Dataset** (Geiger et al., 2013) contains 22 sequences from a Velodyne LiDAR and stereo cameras. Following prior work (Wang et al., 2021a; Liu et al., 2024a), we use the monocular left camera and LiDAR data, with sequences 00–06 for training and 07–10 for testing.

**Argoverse Dataset** (Chang et al., 2019a) provides 113 sequences with LiDAR and stereo images, split into 65/24/24 for training, validation, and testing.

**Evaluation Metrics.** We report (1) average translational RMSE (%) and (2) average rotational RMSE (°/100m) as in PWCLO (Wang et al., 2021a), and also Absolute Trajectory Error (ATE) (Sturm et al., 2012) for SLAM comparison (Engel et al., 2017; Teed & Deng, 2021; Campos et al., 2021).

## 4.2 IMPLEMENTATION DETAILS

**Data Pre-processing.** Following the sparse sampling approach for LiDAR points from (Wang et al., 2022a), we also design a fusion mask that flags LiDAR queries that can interact with image features, given the substantial difference in spatial range between the LiDAR and camera data.

**Hyper-parameters.** We use the Adam optimizer with $\beta_1 = 0.9$ and $\beta_2 = 0.999$. The initial learning rate is 0.001 and decays exponentially every 13 epochs until reaching 0.00001. We use a batch size of 8. The values for $\alpha^i$ across different module outputs are set to 1.6, 0.8, 1.6, and 1.6, respectively. The learnable parameters $k_t$ and $k_q$ are initialized to 0.0 and -2.5, respectively. For each feature

Table 2: **Comparison with traditional visual SLAM (with loop closure) on KITTI 00-10 sequences in ATE[m]↓.**

| Method | 00 ATE | 01 ATE | 02 ATE | 03 ATE | 04 ATE | 05 ATE | 06 ATE | 07 ATE | 08 ATE | 09 ATE | 10 ATE | Mean(00-10) ATE |
|---|---|---|---|---|---|---|---|---|---|---|---|---|
| ORB-SLAM2 (Mur-Artal & Tardós, 2017) | 8.27 | ✗ | 26.86 | 1.21 | 0.77 | 7.91 | 12.54 | 3.44 | 46.81 | 76.50 | 6.61 | - |
| ORB-SLAM3 (Campos et al., 2021) | 6.77 | ✗ | 30.50 | 1.04 | 0.93 | 5.54 | 16.61 | 9.70 | 60.69 | 7.89 | 8.65 | - |
| LDSO(Gao et al., 2018) | 9.32 | 11.68 | 31.98 | 2.85 | 1.22 | 5.10 | 13.55 | 2.96 | 129.02 | 21.64 | 17.36 | 22.42 |
| DROID-SLAM(Teed & Deng, 2021) | 92.11 | 344.62 | ✗ | 2.38 | 1.00 | 118.51 | 62.47 | 21.78 | 161.60 | ✗ | 118.70 | - |
| DPV-SLAM(Lipson et al., 2024a) | 112.80 | 11.50 | 123.53 | 2.50 | 0.81 | 57.80 | 54.86 | 18.77 | 110.49 | 76.66 | 13.65 | 53.03 |
| DPV-SLAM++(Lipson et al., 2024a) | 8.30 | 11.86 | 39.64 | 2.50 | 0.78 | 5.74 | 11.60 | 1.52 | 110.9 | 76.70 | 13.70 | 25.76 |
| MambaVO++(Wang et al., 2025) | 6.19 | 8.04 | 27.73 | 1.94 | 0.59 | 3.05 | 11.79 | 1.7 | 105.42 | 63.24 | 10.51 | 21.84 |
| Ours | 9.28 | 19.72 | 15.23 | 3.51 | 0.76 | 4.90 | 3.61 | 1.00 | 7.74 | 7.27 | 2.73 | 6.91 |

Table 3: **Comparison with traditional visual-LiDAR SLAM on KITTI 00-10 sequences.**

| Method | 00 $t_{rel}$ | 01 $t_{rel}$ | 02 $t_{rel}$ | 03 $t_{rel}$ | 04 $t_{rel}$ | 05 $t_{rel}$ | 06 $t_{rel}$ | 07 $t_{rel}$ | 08 $t_{rel}$ | 09 $t_{rel}$ | 10 $t_{rel}$ | Mean(00-10) $t_{rel}$ |
|---|---|---|---|---|---|---|---|---|---|---|---|---|
| DVL-SLAM (Shin et al., 2020) | 0.93 | 1.47 | 1.11 | 0.92 | 0.67 | 0.82 | 0.92 | 1.26 | 1.32 | 0.66 | 0.70 | 0.98 |
| TVL-SLAM (Chou & Chou, 2021) | 0.59 | ✗ | 0.74 | ✗ | ✗ | 0.32 | 0.32 | 0.36 | 0.88 | 0.64 | ✗ | - |
| HVL-SLAM (Wang et al., 2024a) | 0.75 | 1.86 | 0.81 | 0.87 | 1.09 | 0.57 | 0.70 | 0.80 | 1.08 | 0.71 | 0.81 | 0.91 |
| SDV-LOAM (Yuan et al., 2023) | 0.67 | 0.96 | 0.75 | 0.86 | 0.77 | 0.66 | 0.44 | 0.74 | 1.07 | 0.53 | 0.51 | 0.72 |
| Ours | 0.59 | 0.73 | 0.65 | 0.47 | 0.21 | 0.33 | 0.31 | 0.26 | 0.79 | 0.63 | 0.51 | 0.50 |

level, the number of LiDAR queries is set to 116, 228, 904, and 3600. During training, the KITTI sequences 00-06 are divided into video clips with a duration of $T_C = 60$ frames (6s), with further segmentation into sub-clips of $T_s = 3$ frames (0.3s) for the collective average loss calculation. The maximum history length $T_h$ and the compensation interval $T_g$ are set as 30 and 20 respectively. The value of top-k is set as 100 for the winner-takes-all loss.

## 4.3 QUANTITATIVE RESULTS

**Comparison with Deep Odometry Methods on KITTI.** We compare our model with recent learning-based visual odometry (VO), LiDAR odometry (LO), and multi-modal methods. Following (Wang et al., 2021a), our model is trained on sequences 00-06. The main results on the KITTI dataset are presented in Table 1, demonstrating that ST-VLO surpasses these methods on most sequences. Specifically, compared to deep visual odometry methods (Zhan et al., 2021; Cho & Kim, 2023), our method has lower estimation errors on sequences 07-10 with a 75% $t_{rel}$ and 60% $r_{rel}$ reduction, respectively. It is worth mentioning that while these VO methods are generally trained on a larger dataset (00-08), ST-VLO still achieves significantly better results. Compared to recent state-of-the-art LO methods, ST-VLO exceeds EfficientLO (Wang et al., 2022a), DSLO (Zhang et al., 2024a), and LAGLO (Tang et al., 2025) on most sequences. Compared to multi-modal odometry methods, ST-VLO achieves a reduction in average translation error by 19% and average rotation error by 22% compared to the state-of-the-art visual-LiDAR odometry method DVLO (Liu et al., 2024a), highlighting ST-VLO's capabilities in effective spatial and temporal modeling.

**Comparison with Traditional SLAMs on KITTI.** To further validate the efficacy of our method in long-term temporal modeling for drift mitigation, we compare ST-VLO with recent traditional visual SLAM systems (Table 2) and traditional visual-LiDAR SLAM systems (Table 3).

ST-VLO outperforms these methods on most sequences in terms of Absolute Trajectory Error (ATE) and relative translation errors, consistently demonstrating its effectiveness in mitigating drift. It is worth noting that these SLAM systems rely on global optimization and loop closure across a sequence of frames to reduce accumulative errors, whereas our method leverages Temporal Drift Composition to backtrack previously observed frames and compensate for accumulative errors, without requiring explicit mapping or loop closure. This property makes ST-VLO more adaptable to sequences of arbitrary lengths in various driving scenes.

**Comparison Results on Argoverse.** To evaluate our method's generalization capability, we perform experiments on the Argoverse dataset (Chang et al., 2019a). Following the protocol in (Zhang et al., 2024a), we employ Absolute Trajectory Error (ATE) and Relative Pose Error (RPE) as evaluation metrics. ST-VLO is trained and evaluated using the official Argoverse training/testing split. As shown in Table 4, ST-VLO surpasses four traditional geometry-based odometry methods (Shan & Englot, 2018; Behley & Stachniss, 2018; Dellenbach et al., 2021; Qin & Cao) without mapping for

Table 4: **Experiments on the Argoverse dataset (Chang et al., 2019a).**

| Method | LeGO-LOAM (Shan & Englot, 2018) | SUMA (Behley & Stachniss, 2018) | PyLiDAR (Dellenbach et al., 2021) | A-LOAM (Qin & Cao) | DSLO (Zhang et al., 2024a) | DVLO (Liu et al., 2024a) | DVLO4D (Liu et al., 2025b) | ST-VLO (Ours) |
|---|---|---|---|---|---|---|---|---|
| ATE | 4.537 | 3.663 | 6.900 | 4.138 | 0.111 | 0.103 | 0.089 | **0.073** |
| RPE | 0.110 | 0.039 | 0.109 | 0.066 | 0.027 | 0.026 | 0.025 | **0.021** |

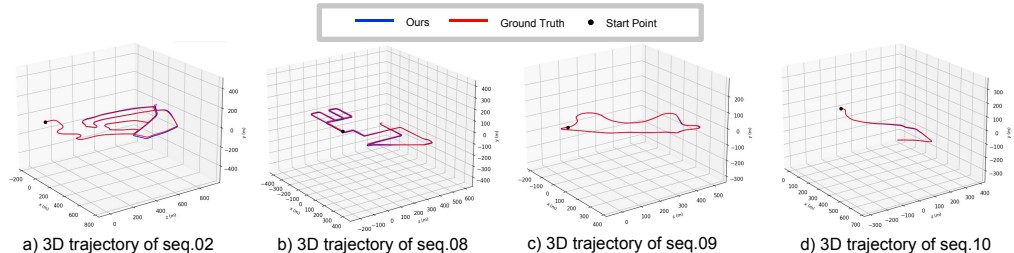

| a) 3D trajectory of seq.02 | b) 3D trajectory of seq.08 | c) 3D trajectory of seq.09 | d) 3D trajectory of seq.10 |

Figure 4: **Trajectory of our estimated pose on KITTI.**

a fair comparison and also the recent state-of-the-art learning-based method DSLO (Zhang et al., 2024a) by 18% in ATE and 16% in RPE.

### 4.4 QUALITATIVE RESULTS

We provide 3D trajectory visualizations derived from our estimated poses in Fig. 4. The results illustrate that our odometry approach closely aligns with the ground truth trajectory. Additionally, we perform experiments to compare the trajectory accuracy and estimation errors between our method and the classical LOAM method (Zhang & Singh, 2014) in Fig. 5. Notably, even though our odometry only serves as the front end and does not include mapping, it achieves superior localization accuracy compared to LOAM with mapping.

Moreover, we plot the top-k query points in Fig. 6 to showcase the high-salient feature regions. Most points are located in regions of static objects such as buildings, and static cars, but fewer points are seen on dynamic objects such as moving cars and pedestrians. This is because dynamic objects introduce inconsistent motions, undermining the accuracy of localization (Liu et al., 2023b), while static objects provide a more reliable reference to estimate the ego-vehicle's motions.

### 4.5 LATENCY ANALYSIS

As presented in Table 5, we compare ST-VLO latency with other multi-modal odometry methods on a single NVIDIA 4090 GPU. Efficiency is critical for real-time SLAM, as KITTI LiDAR is sampled at 10 Hz. Many existing multi-modal approaches (Wang et al., 2021b; Shu & Luo, 2022; Liu et al., 2024a) struggle to meet the 100 ms real-time threshold. In contrast, our method achieves an inference latency of 74 ms, showcasing its potential for real-time applications.

Table 5: **Average inference time of different methods on the sequence 07-10 of KITTI dataset.**

| Method | PL-LOAM (Huang et al., 2020a) | DV-LOAM (Wang et al., 2021b) | Shu *et al.* (Shu & Luo, 2022) | DVLO (Liu et al., 2024a) | ST-VLO (Ours) |
|---|---|---|---|---|---|
| Time (ms) | 200 | 167 | 100 | 99 | **74** |

### 4.6 ABLATION STUDIES

We conduct various ablation studies to demonstrate the effectiveness of each component in our proposed ST-VLO network on the KITTI dataset. They are a) the unified *MMG* for both spatial and temporal inputs, b) the *Temporal Drift Compensation*, c) the *Key Points-Aware Auxiliary Loss*, and d) integrating all components. As shown in Table 6, overall, removing any of the proposed components leads to inferior performance in mean translation and rotation errors, highlighting the significance of each design choice in our architecture. This is most profound for the unified MMG, which enhances performance by creating a unified feature space for harnessing spatial and temporal information from the visual-LiDAR data. Second, Temporal Drift Compensation improves results by iteratively correcting cumulative errors, which is also evidently visible as shown in Fig 7. Last, the Key Points-

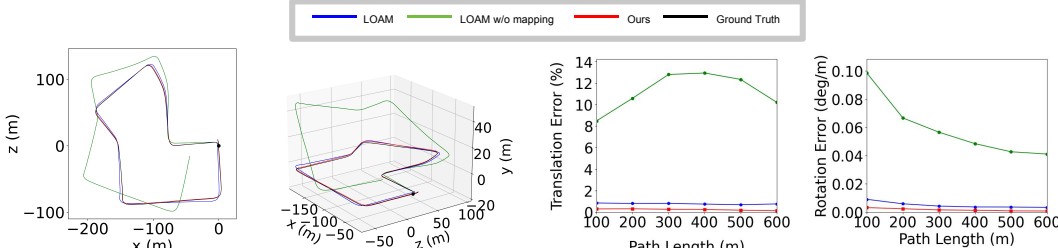

Figure 5: **Trajectory and error comparison on KITTI (seq. 07).**

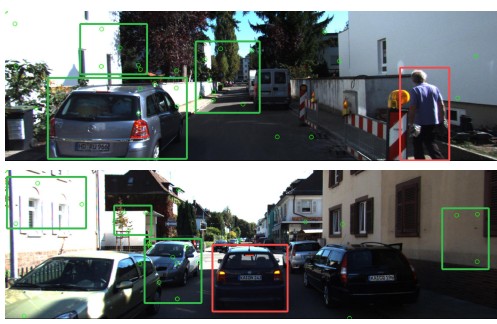

Figure 6: **Visualization of the Keypoints-Aware Auxiliary Loss.** Green points mark the top-k queries with the smallest errors, concentrated in static regions (green boxes) and fewer in dynamic objects (red boxes).

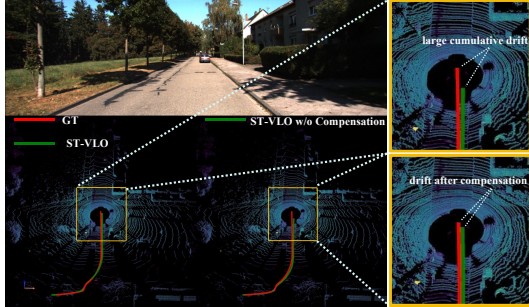

Figure 7: **Comparison of with vs. without the Temporal Drift Compensation.** Our designed *Temporal Drift Compensation* module can adjust the estimation drifts by iteratively calculating the accumulated errors.

Aware Auxiliary Loss strengthens the learning of robust feature regions, which contributes to an overall better performance by 12%.

We explore fusion strategies to evaluate deformable Mamba, which achieves the highest accuracy and second fastest latency (74 ms), outperforming attention-based (Liu et al., 2021b), cluster-based (Liu et al., 2024a), and deformable attention-based (Zhu et al., 2021) methods (Table 7).

Additional ablation studies on query numbers in the top-k winner-takes-all loss, frame lengths $T_h$ and $T_g$, and extra visualizations are provided in the supplementary materials.

Table 6: **Ablation study of main design choices.** The best results are in **bold**.

| Method | 07 | | 08 | | 09 | | 10 | | Mean | |
|---|---|---|---|---|---|---|---|---|---|---|
| | $t_{rel}$ | $r_{rel}$ | $t_{rel}$ | $r_{rel}$ | $t_{rel}$ | $r_{rel}$ | $t_{rel}$ | $r_{rel}$ | $t_{rel}$ | $r_{rel}$ |
| (a) w/o Unified MMG | 0.33 | 0.35 | 0.91 | 0.44 | 0.69 | 0.34 | 0.87 | 0.38 | 0.70 | 0.38 |
| (b) w/o Compensation | 0.35 | 0.32 | 0.83 | 0.39 | 0.74 | 0.33 | 0.88 | 0.40 | 0.70 | 0.36 |
| (c) w/o Auxiliary Loss | 0.31 | 0.29 | 0.87 | 0.43 | 0.71 | 0.34 | 0.79 | 0.44 | 0.67 | 0.38 |
| (d) ST-VLO | **0.27** | **0.21** | **0.79** | **0.28** | **0.63** | **0.28** | **0.65** | **0.39** | **0.59** | **0.29** |

Table 7: **Ablation study of multi-modal fusion strategies.** The best results are in **bold**.

| Method | 07 | | 08 | | 09 | | 10 | | Mean | | Latency |
|---|---|---|---|---|---|---|---|---|---|---|---|
| | $t_{rel}$ | $r_{rel}$ | $t_{rel}$ | $r_{rel}$ | $t_{rel}$ | $r_{rel}$ | $t_{rel}$ | $r_{rel}$ | $t_{rel}$ | $r_{rel}$ | (ms) |
| CNN (Huang et al., 2020b) | 0.39 | 0.34 | 1.11 | 0.49 | 0.81 | 0.44 | 0.91 | 0.52 | 0.81 | 0.45 | **71** |
| Clustering (Liu et al., 2024a) | 0.34 | 0.37 | 0.93 | 0.41 | 0.78 | 0.43 | 0.92 | 0.43 | 0.74 | 0.41 | 87 |
| Attention (Liu et al., 2021b) | 0.36 | 0.28 | 0.97 | 0.41 | 1.01 | 0.49 | 0.85 | 0.48 | 0.80 | 0.42 | 171 |
| Deform DETR (Zhu et al., 2021) | 0.32 | 0.35 | 0.87 | 0.37 | 0.71 | 0.32 | 0.69 | 0.34 | 0.65 | 0.35 | 76 |
| Deform Mamba | **0.27** | **0.21** | **0.79** | **0.28** | **0.63** | **0.28** | **0.65** | **0.39** | **0.59** | **0.29** | 74 |

## 5 CONCLUSION

In this work, we introduced ST-VLO, a visual-LiDAR odometry framework designed to address the challenges of accumulative drifts and computational efficiency in the odometry task. By integrating a unified MMG architecture, ST-VLO combines visual and LiDAR data, enabling robust feature fusion while capturing essential spatiotemporal dependencies. Our model leverages a Temporal Drift Compensation module to minimize the cumulative drift. Moreover, a Keypoint-Aware Auxiliary Loss is proposed to enhance spatial feature representations in high-salient regions. Extensive experiments demonstrate that ST-VLO achieves state-of-the-art performance in both KITTI and Argoverse datasets. Notably, ST-VLO is efficient with an inference speed of over 10 Hz, which has the potential for real-time application in autonomous driving scenes.

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

APPENDIX

# A OVERVIEW

The supplementary materials are structured as follows:

- We give more detailed illustrations about the network architecture of the MMG module in Section B;

- More experimental results about the ablation studies are provided in Section C.

- We display more visualization results in Section D.

- Section E discloses the limited and strictly assistive usage of a large language model (LLM) during manuscript polishing.

- Also, a video demo of real-world driving scenes is appended to the supplement materials.

# B ARCHITECTURE OF THE MMG MODULE

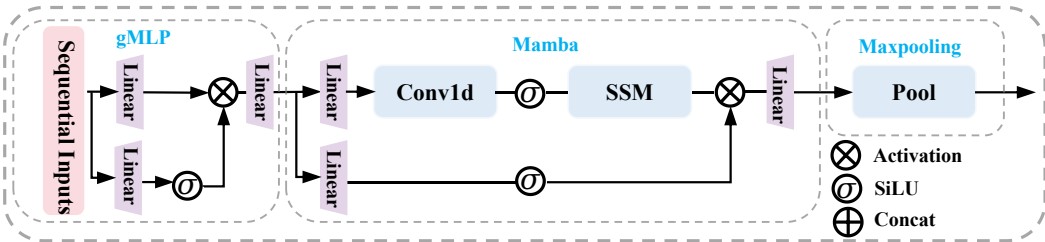

Figure 8: **The detailed network structure of the MMG module.**

Fig. 8 shows the detailed network structure of MMG (MaxPooling, Mamba (Gu & Dao, 2024), and gMLP (Liu et al., 2021a)). MMG harmonizes diverse input representations across different modalities and temporal dimensions: The gMLP encodes sequential information into a unified feature space, providing a foundation for learning from varied data sources. Mamba then establishes temporal interactions within the sequences, effectively capturing long-term dependencies across frames. Finally, MaxPooling condenses the sequence into a single, unified representation, preserving essential information in a compact form.

In this paper, we incorporate the Mamba block proposed in (Gu & Dao, 2024) into our method for processing image and LiDAR data due to its excellent performance and speed. Mamba (Gu & Dao, 2024) is designed for linear-time sequence modeling using structured state space sequence models (SSMs) (Gu et al., 2022). These models are extended to selectively propagate or forget information along the temporal dimension based on the current input token. Specifically, let $X$ denote the input features derived from the image and point cloud data processed by the gMLP layer. These features then serve as the input tokens for the Mamba block:

$$\hat{X} = \text{LN}(X), \tag{7}$$

$$\overline{X} = \sigma(\text{Conv1D}(\text{Linear}(\hat{X}))), \tag{8}$$

$$\hat{X} = \sigma(\text{Linear}(\hat{X})), \tag{9}$$

$$Y = \text{Linear}(\text{SSM}(\overline{X})) \odot \hat{X} + X, \tag{10}$$

where $\sigma$ denotes the SiLU activation function (Hendrycks & Gimpel, 2016), Conv1D denotes a 1D convolution layer, LN denotes the linear normalization, and SSM is the standard selective state space model proposed by Gu & Dao (2024). The output $Y$ denotes the temporally encoded features for the consequent Maxpooling Layer.

## C ADDITIONAL ABLATION STUDIES

We provide more ablation and generalization studies on KITTI (Geiger et al., 2013), Argoverse (Chang et al., 2019a), and Hilti SLAM'22/'23 (Nair et al., 2024; Helmberger et al., 2022) datasets to analyze different settings of our proposed method.

### C.1 GENERALIZATION ABILITY

**Generalization to Indoor 6-DoF Scenarios.** We evaluate on the Hilti SLAM'22/'23 (Nair et al., 2024; Helmberger et al., 2022) open 6-DoF dataset, which stresses odometry with strong elevation changes, unconstrained rotations, long corridors, stairwells, and low-light rooms. Across these diverse scenes, our method consistently yields trajectories that adhere closely to ground truth (Fig.9) and outperforms SDV-LOAM (Yuan et al., 2023) on the vast majority of sequences (Table 8). We observe fewer drift accumulations at vertical transitions and better stability in texture-poor or dim environments, indicating that the unified spatio-temporal correlation and deformable multi-modal fusion provide stronger constraints than purely geometric pipelines. Notably, these gains are obtained without explicit loop closure or global mapping and at real-time latency, underscoring practicality for indoor robotics.

Table 8: **Evaluation on 6-DoF scenes (Hilti SLAM'22/'23). Lower is better.**

| Method | Con. gr | Con. m | Con. st | Long corr. | Cupola | Low. gall. | Attic→up. gall. | Floor 0 |
|---|---|---|---|---|---|---|---|---|
| SDV-LOAM (Yuan et al., 2023) | 25.1 | **12.6** | 9.2 | 19.5 | 9.3 | 11.2 | **4.6** | 4.6 |
| Ours | **20.1** | 13.3 | **8.2** | **17.3** | **7.6** | **9.3** | 5.1 | **4.0** |

| Method | Floor 1 | Floor 2 | Base. | Stairs | P. 3×flr | L. rm. | L. rm. (dark) | Mean |
|---|---|---|---|---|---|---|---|---|
| SDV-LOAM (Yuan et al., 2023) | 8.0 | 7.9 | 6.2 | 9.0 | 20.0 | **16.8** | 15.0 | 11.9 |
| Ours | **6.3** | **3.3** | **2.9** | **5.3** | **15.2** | 19.1 | **9.9** | **9.8** |

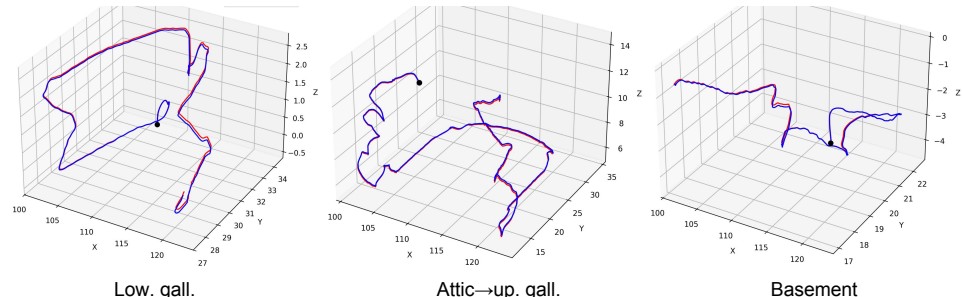

| Low. gall. | Attic→up. gall. | Basement |

Figure 9: **Estimated 6-DoF pose trajectories on the Hilti dataset.**

**Cross-Dataset Generalization.** Trained on KITTI and tested on Argoverse, ST-VLO degrades much less than DVLO: relative to in-domain training on Argoverse, ST-VLO increases by +32.9% (ATE) and +19.0% (RPE), whereas DVLO increases by +81.6% and +34.6%, respectively. Under this cross-domain setting, ST-VLO still outperforms DVLO (ATE: 0.097 vs. 0.187; RPE: 0.025 vs. 0.035), indicating stronger generalization ability of our method.

Table 9: **Pose errors on Argoverse (cross-dataset from KITTI).** Percentages (in red) denote the relative *increase* vs. each method's in-domain result (trained on Argoverse); lower is better.

| Method | Training | Testing | ATE (m) | RPE (m) |
|---|---|---|---|---|
| DVLO (Liu et al., 2024a) | Argoverse | Argoverse | 0.103 | 0.026 |
| ST-VLO | Argoverse | Argoverse | 0.073 | 0.021 |
| DVLO (Liu et al., 2024a) | KITTI | Argoverse | 0.187 (+81.6%) | 0.035 (+34.6%) |
| ST-VLO | KITTI | Argoverse | 0.097 (+32.9%) | 0.025 (+19.0%) |

### C.2 PERFORMANCE ON COMPLEX MOTION

We further probe two challenging regimes on KITTI—high dynamics (fast ego motion) and high rotation rate—where pairwise or short-horizon models typically struggle. As summarized in Table10, introducing long-range temporal modeling and drift compensation stabilizes pose estimates during rapid accelerations, sharp turns, and heading changes: trajectories remain well aligned, corners are

preserved rather than over-smoothed, and heading oscillations are markedly reduced. Qualitative inspection reveals fewer slip events and faster recovery in feature-sparse spans, supporting our claim that robust temporal cues—not merely stronger per-frame features—are critical for reliable odometry under aggressive maneuvers.

Table 10: **Performance on high-dynamics and high-rotation scenarios of KITTI.**

| Method | High Dynamics | | High Rotation Rate | |
|---|---|---|---|---|
| | $t_{\mathrm{rel}}$ | $r_{\mathrm{rel}}$ | $t_{\mathrm{rel}}$ | $r_{\mathrm{rel}}$ |
| DVLO (Liu et al., 2024a) | 0.91 | 0.39 | 1.63 | 0.69 |
| ST-VLO | **0.67** | **0.20** | **0.79** | **0.28** |

### C.3 ABLATION STUSIES

**Query Numbers in the Top-k Winner-takes-all Loss.** As shown in Table 11, by increasing the number of queries from 50 to 200, the performance on most of the sequences starts to decrease after $k = 100$. Hence, we opt $k = 100$ for the top-k winner-takes-all of the keypoint-aware auxiliary loss.

Table 11: **The impact of varying the number of queries (top-k) for the winner-takes-all loss.**

| top-k | 07 | | 08 | | 09 | | 10 | | Mean | |
|---|---|---|---|---|---|---|---|---|---|---|
| | $t_{rel}$ | $r_{rel}$ | $t_{rel}$ | $r_{rel}$ | $t_{rel}$ | $r_{rel}$ | $t_{rel}$ | $r_{rel}$ | $t_{rel}$ | $r_{rel}$ |
| 50 | 0.34 | 0.23 | 0.83 | 0.30 | 0.68 | 0.30 | 0.69 | 0.43 | 0.64 | 0.32 |
| 100 | **0.26** | **0.23** | 0.79 | 0.28 | **0.63** | **0.28** | **0.65** | 0.39 | **0.59** | **0.29** |
| 200 | 0.38 | 0.40 | **0.77** | **0.25** | 0.69 | 0.33 | 0.79 | 0.42 | 0.66 | 0.35 |

**Varying Frame Lengths.** We also analyze the impact of frame lengths for the sub-clip $T_s$, the maximum history frame length $T_h$, and the compensation interval $T_g$. As shown in Table 12, aggregating losses (Section 3.3 in the main paper) multiple frames ($T_s = 3$) leads to a better performance than the single-frame ($T_s = 1$) losses. By varying the maximum history frame length $T_h$ from 15 to 45, the performance first increases and then starts to decrease after increasing $T_h$ to 30. Similarly, the compensation interval $T_g = 20$ yields the best performance among the other frame lengths. Overall, when $T_s = 3$, $T_h = 30$, and $T_g = 20$, our method achieves the best performance.

Table 12: **The impact of varying the sub-clip length $T_s$, the maximum history frame length $T_h$, and the compensation interval $T_g$.** The best results are **bold**.

| $T_s$ | $T_h$ | $T_g$ | 07 | | 08 | | 09 | | 10 | | Mean | |
|---|---|---|---|---|---|---|---|---|---|---|---|---|
| | | | $t_{rel}$ | $r_{rel}$ | $t_{rel}$ | $r_{rel}$ | $t_{rel}$ | $r_{rel}$ | $t_{rel}$ | $r_{rel}$ | $t_{rel}$ | $r_{rel}$ |
| 1 | 30 | 20 | 0.33 | 0.29 | 0.91 | 0.38 | 0.85 | 0.43 | 0.89 | 0.53 | 0.75 | 0.41 |
| 3 | 15 | 20 | 0.29 | 0.31 | **0.76** | 0.29 | 0.67 | 0.31 | 0.70 | **0.35** | 0.61 | 0.32 |
| 3 | 30 | 20 | **0.26** | **0.23** | 0.79 | **0.28** | 0.63 | **0.28** | **0.65** | 0.39 | **0.59** | **0.29** |
| 3 | 45 | 20 | 0.30 | 0.33 | 0.83 | 0.29 | 0.67 | 0.32 | 0.70 | 0.41 | 0.63 | 0.34 |
| 3 | 30 | 10 | 0.35 | 0.39 | 0.87 | 0.35 | 0.73 | 0.37 | 0.71 | 0.43 | 0.67 | 0.39 |
| 3 | 30 | 30 | 0.31 | 0.32 | 0.87 | 0.32 | **0.61** | 0.31 | 0.75 | 0.45 | 0.64 | 0.35 |

## D VISUALIZATION OF THE RESULTS

### D.1 2D & 3D TRAJECTORY VISUALIZATION

We display the 2D and 3D trajectories on all the evaluation sequences 00-10 of the KITTI dataset respectively in Fig. 10 and Fig. 11. As shown in these figures, our estimated trajectories consistently overlap with the ground truth ones well, which demonstrates the superiority of our proposed odometry method.

### D.2 KEYPOINTS-AWARE AUXILIARY LOSS

In Fig. 12 and Fig. 13, we also show more visualizations of the selected top-k keypoints with minimal error relative to the ground truth pose in our designed keypoints-aware auxiliary loss. From these figures, most keypoints are located in the regions of static objects, such as buildings, trees, and unmoving/parked cars, because these static objects contribute the most to a consistent ego-motion estimation. In contrast, there are very few selected keypoints located in the regions of dynamic objects since, e.g., moving cars or pedestrians can introduce different motion patterns that degrade the motion consistency. Our carefully designed top-k winner-takes-all strategy avoids this undesirable effect in the keypoints-aware auxiliary loss.

## E LLM USAGE STATEMENT

A large language model (ChatGPT) was used only for limited editing support during manuscript preparation. Its role was restricted to: (i) checking spelling and grammar; (ii) light phrasing and wording adjustments to improve readability without changing technical content, methodology, analyses, or conclusions; and (iii) occasional condensation of repetitive sentences and suggestions for consistent formatting. The LLM did not participate in research ideation, problem formulation, method design, experiments, data processing, result analysis, drafting of technical material, or drawing conclusions. It is not an author and bears no responsibility for the manuscript's content.

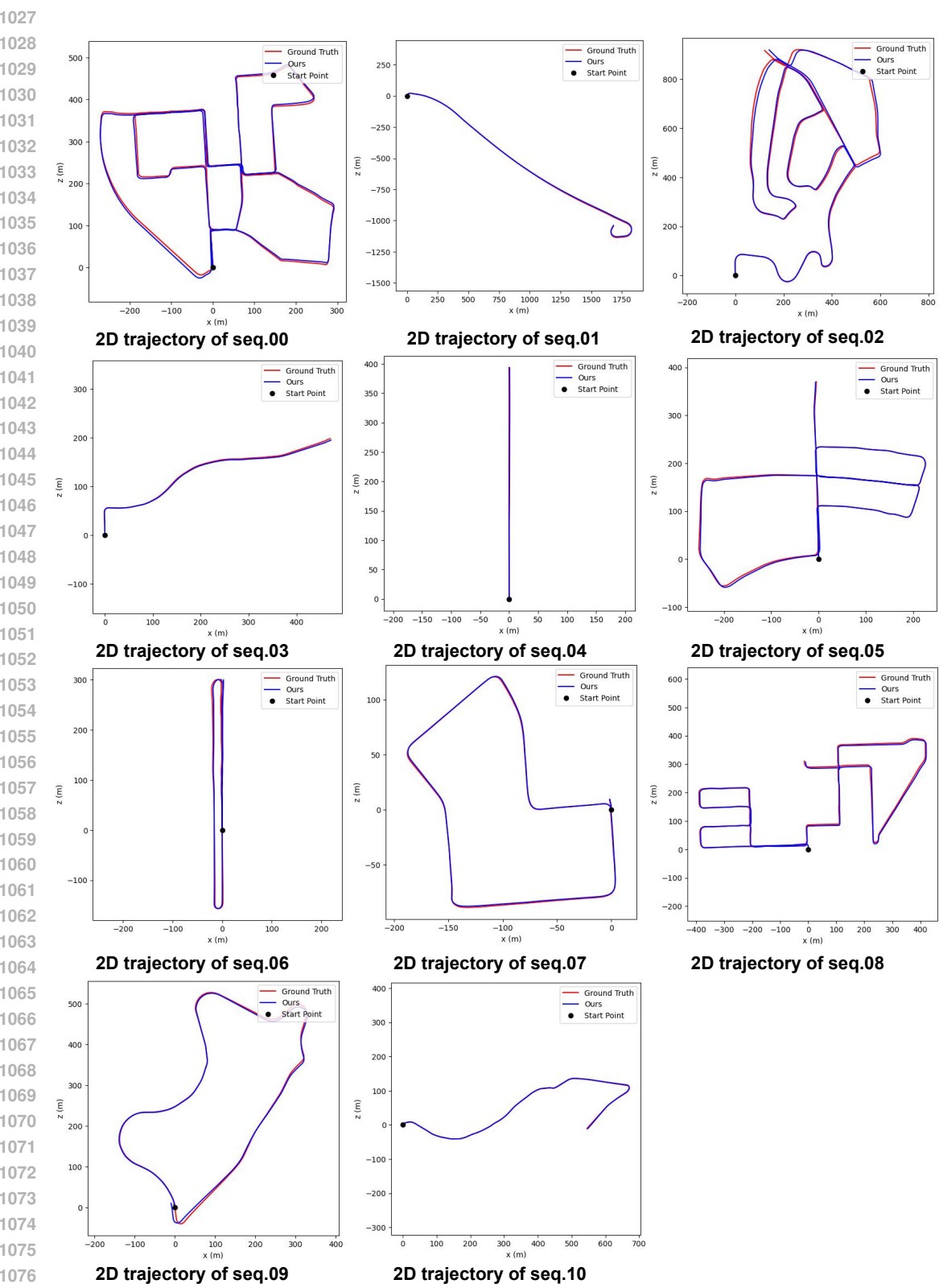

Figure 10: **The 2D trajectories of ground-truth pose and our estimated pose.** Comprehensive 2D trajectory results are shown here on 00-10 sequences of the KITTI dataset.

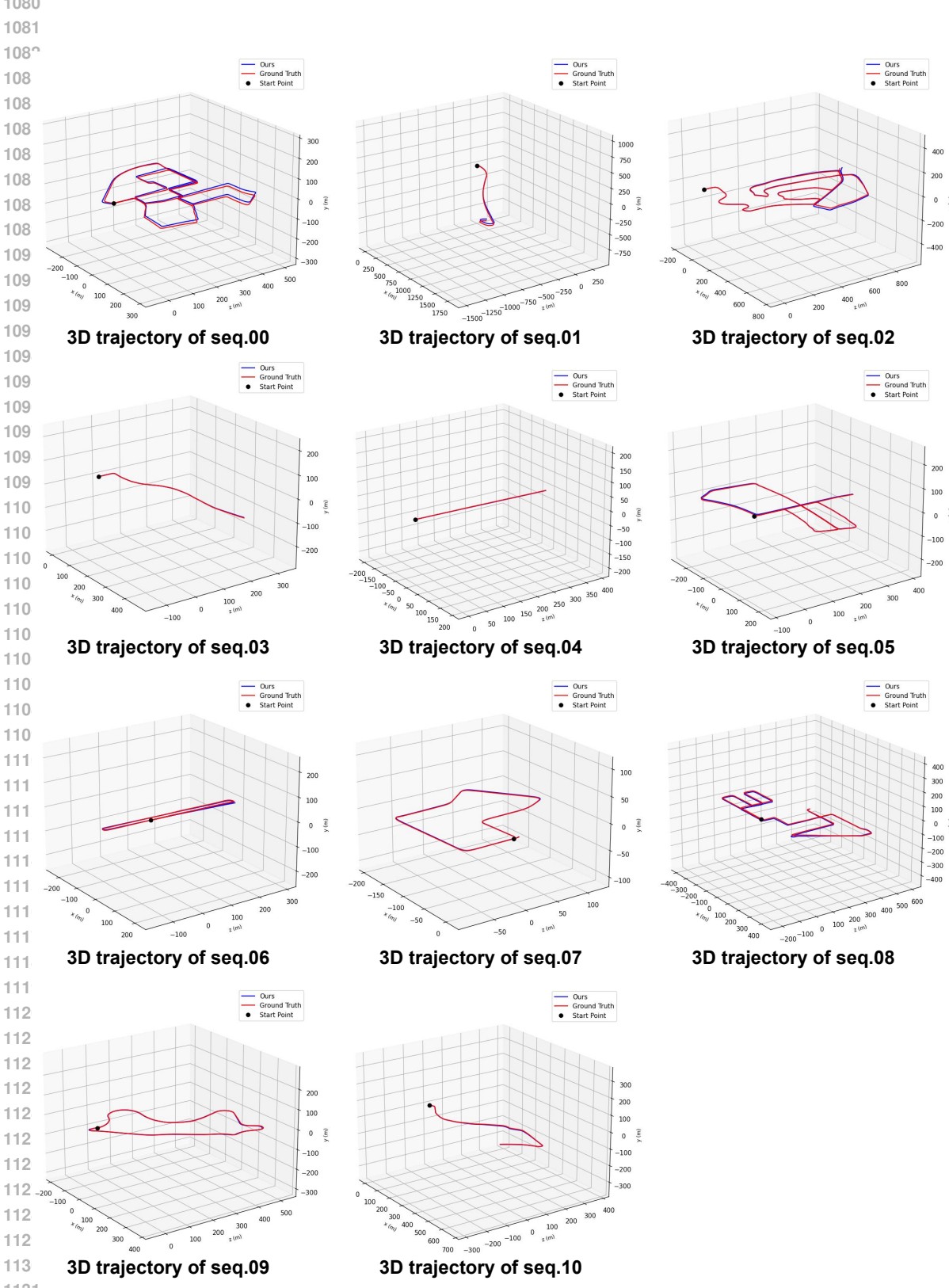

Figure 11: **The 3D trajectories of ground-truth pose and our estimated pose.** Comprehensive 3D trajectory results are shown here on 00-10 sequences of the KITTI dataset.

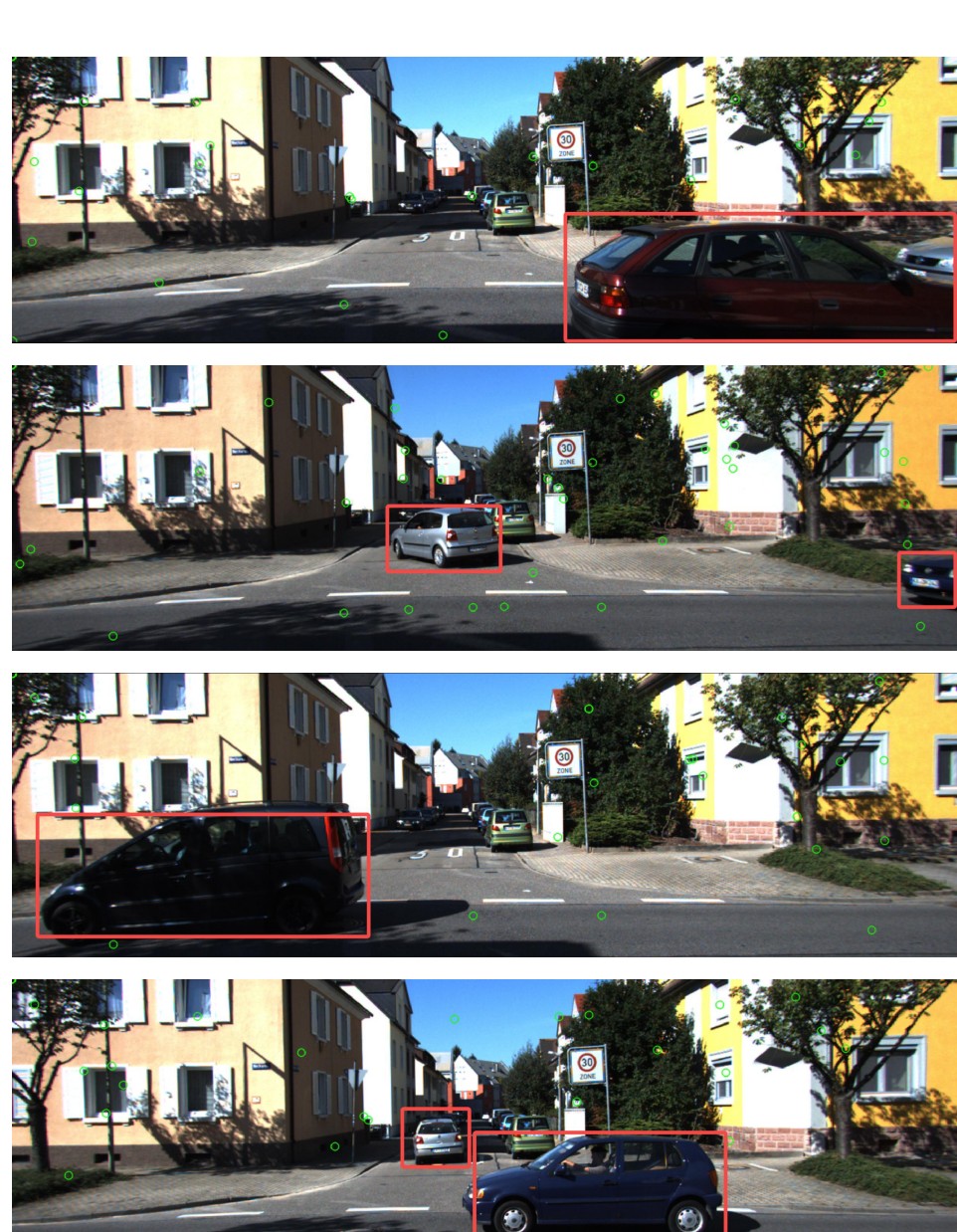

Figure 12: **Visualization of the Keypoints-Aware Auxiliary Loss (1).** Green points indicate the top-k queries with minimal error relative to the ground truth pose, showing that they are primarily located in regions associated with static objects, while less focus on dynamic objects (red boxes).

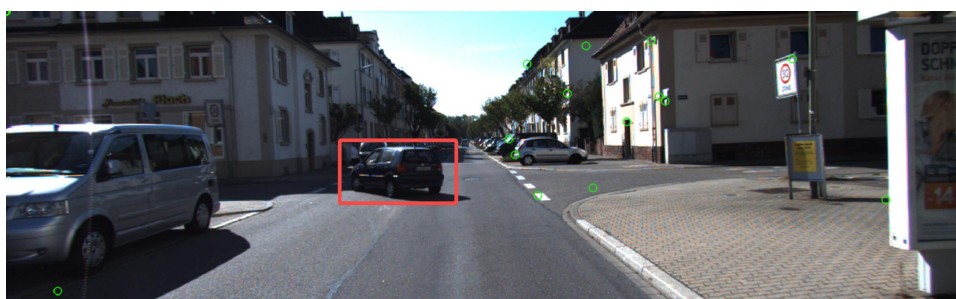

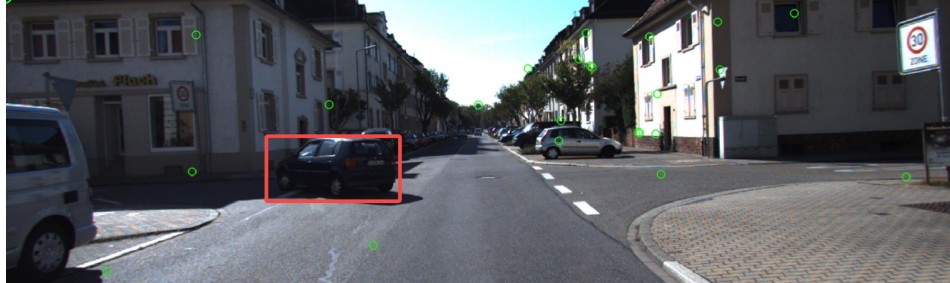

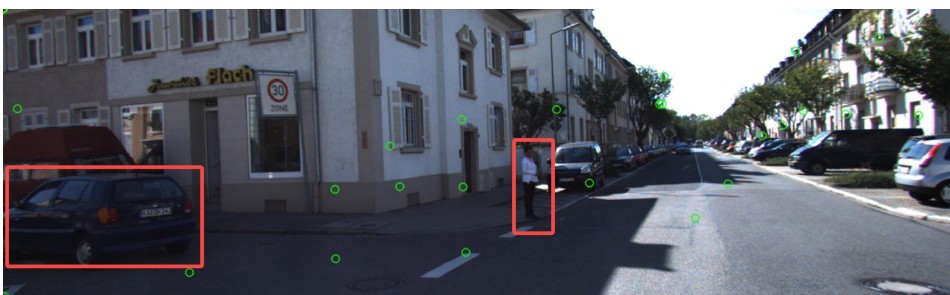

Figure 13: **Visualization of the Keypoints-Aware Auxiliary Loss (2).** Green points indicate the top-k queries with minimal error relative to the ground truth pose, showing that they are primarily located in regions associated with static objects, while less focus on dynamic objects (red boxes).

