# OpenReview forum: "ST-VLO: Unified Spatio-Temporal Correlation for Visual-LiDAR Odometry with Temporal Drift Compensation"
_ICLR.cc/2026/Conference — ICLR 2026 Conference Withdrawn Submission_

### Official Review · Reviewer_vUSs · 2025-10-26

**Soundness:** 3
**Presentation:** 3
**Contribution:** 3
**Rating:** 6
**Confidence:** 5

**Summary:**

This paper introduces ST-VLO, a novel visual-LiDAR odometry framework that unifies spatial and temporal feature modeling to achieve accurate and low-drift motion estimation in dynamic environments. Unlike conventional pairwise odometry methods that only process consecutive frame pairs, ST-VLO leverages multi-frame temporal information through a Mamba-based spatio-temporal fusion module called MMG (MaxPooling-Mamba-gMLP). This module jointly models correlations across visual and LiDAR modalities over time, providing richer motion context.
To further mitigate cumulative localization errors, the authors propose a Temporal Drift Compensation mechanism that iteratively corrects pose predictions using residuals from historical frames. In addition, a Keypoint-Aware Auxiliary Loss emphasizes static, high-salience regions to enhance feature robustness and reduce the influence of dynamic objects.
Experiments on the KITTI and Argoverse benchmarks demonstrate that ST-VLO achieves state-of-the-art performance, reducing translation and rotation errors by about 19% and 22%, respectively, compared to the best prior deep visual-LiDAR odometry methods. The framework also runs efficiently (≈74 ms per frame), showing strong potential for real-time autonomous driving applications.

**Strengths:**

1. The proposed MMG (MaxPooling-Mamba-gMLP) module is an interesting design that harmonizes visual and LiDAR features across space and time. The formulation is well motivated by recent developments in SSMs and avoids the high computational burden of Transformers.
2. The compensation mechanism that iteratively refines poses over historical windows is original and practically valuable for odometry tasks. It provides an elegant, differentiable alternative to traditional loop closure.
3. The real-time efficiency is remarkable. Achieving 74 ms inference per frame on a 4090 GPU (≈13 Hz) demonstrates strong engineering quality and practical relevance.

**Weaknesses:**

1. It is an interesting endeavor to apply Mamba to odometry estimation. However, while Mamba-based modeling and temporal compensation are well-executed, each component is adapted from existing paradigms (Deformable DETR, PWC-LO, Mamba SSM). The overall contribution is more of an incremental integration than a fundamentally new principle.

2. Table 6 evaluates the three main modules but lacks quantitative breakdowns for parameters like temporal window sizes (Th, Tg), compensation frequency, or loss weighting. These details are essential for understanding the stability and generality of the approach. Moreover, the performance gaps between different ablation variants are not very appreciable.

3. The pipeline description is dense and formula-heavy; key intuitions behind design decisions (e.g., why use MaxPooling in MMG, why gMLP before Mamba) are missing. The paper would benefit from schematic clarity or pseudocode.

4. Figure 6 highlights static regions for keypoints but doesn’t quantify how this improves robustness in dynamic scenes (e.g., removing moving object influence).

5. The discussion on unsupervised methods is missing. I have listed a few for reference:
[1] Generalizing Unsupervised Lidar Odometry Model from Normal to Snowy Weather Conditions
[2] Self-supervised learning of lidar odometry for robotic applications
[3] Robust selfsupervised lidar odometry via representative structure discovery and 3d inherent error modeling,
[4] Hpplo-net: Unsupervised lidar odometry using a hierarchical point-to-plane solve

**Questions:**

1. How is the history length (Th) chosen? How sensitive is performance to this parameter?
2. How would ST-VLO handle large temporal gaps or asynchronous visual/LiDAR frequencies?
3. Does the system require calibration refinement during training, or is extrinsic calibration fixed?

---

### Official Review · Reviewer_no3p · 2025-10-27

**Soundness:** 2
**Presentation:** 2
**Contribution:** 2
**Rating:** 2
**Confidence:** 4

**Summary:**

The paper proposes ST-VLO, a unified spatio-temporal visual-LiDAR odometry framework. The method follows closely the multi-modal architecture of DVLO4D (Liu et al, 2025b) and introduces three components: (i) the Mamba-based multi-modal, spatio-temporal fusion module; (ii) the Temporal Drift Compensation (TDC) that learns residual corrections; (iii) the Keypoint-Aware Auxiliary loss that selects the top-K "best" query predictions. The approach is evaluated on KITTI and Argoverse datasets and achieves slightly improved pose estimates over DVLO4D, with several qualitative visualizations and ablations.

**Strengths:**

1. The overall system design is clear, and the results on outdoor benchmarks are competitive.
2. The inclusion of a drift-compensation mechanism is reasonable and could improve long-term consistency.

**Weaknesses:**

1. The proposed architecture closely mirrors DVLO4D in both design and objectives, which utilizes multi-modal and temporal information to increase pose accuracy and robustness. In particular, there is a close resemblance between the proposed method and DVLO4D in the query-based feature fusion across image and LiDAR modalities, temporal aggregation via memory bank, and iterative pose refinement. The paper does not provide a controlled comparison between the existing DVLO4D modules and proposed blocks, underlying the motivation and gains behind the change.

2. Table 12 provides a quantitative analysis of the selection of temporal window length supporting the claim in lines 172-173 that Mamba-based MMG captures long-range dependencies with $T_h = 30$. However, DVLO4D already reports the same optimal window length using a transformer backbone. This indicates that the gain comes from using more temporal context rather than any architectural advantage of Mamba.

3. TDC computes residual corrections over cumulative poses, effectively acting as iterative residual alignment. This is conceptually similar to DVLO4D's iterative refinement, yet no side-by-side comparison is provided. Without such an ablation, it is impossible to evaluate the claimed contribution.

4. The description of the Keypoint-Aware Auxiliary loss is unclear. Queries are ranked by ground-truth pose error, and the top-K are optimized; however, no mapping from queries to spatial keypoints or any keypoint sampling is defined. During inference, ground truth data is unavailable, so it is unclear how "keypoints" are determined.

5. In Figures 6, 12, and 13, the selected keypoints are shown to lie on static objects, but this seems emergent rather than designed, since no clear sampling is defined (see W4). The mechanism appears, therefore, heuristic and lacks conceptual rigor.

**Questions:**

1. Since the model has been trained on KITTI sequences 00-06, it is not informative to include the results on those sequences in Tables 1-3.
2. Considering the known, dataset-specific calibration between LiDAR and camera, how is it mitigated for the generalization claims in experimental lines 900 - 904?

---

### Official Review · Reviewer_Pupc · 2025-10-31

**Soundness:** 3
**Presentation:** 3
**Contribution:** 2
**Rating:** 4
**Confidence:** 4

**Summary:**

This paper proposes ST-VLO, a visual-LiDAR odometry framework that addresses cumulative drift in autonomous driving scenarios.

**Strengths:**

1. Well-motivated problem: Addressing cumulative drift in multi-frame visual-LiDAR odometry is practically important for autonomous driving applications.
2. Strong empirical results: Achieves 19% t_rel and 22% r_rel improvements over DVLO on KITTI, with competitive real-time performance (74ms inference).
3. Thorough ablation studies: The supplementary material provides detailed ablations on fusion strategies, frame lengths, and top-k values, showing careful empirical investigation.
4. Unified multi-modal temporal modeling: The MMG architecture provides a cohesive framework for processing heterogeneous sensor data across temporal dimensions

**Weaknesses:**

1. Insufficient justification for Mamba over Transformer: The paper claims Mamba provides "efficient" modeling but lacks theoretical complexity analysis (O(N) vs O(N²)) and direct experimental comparisons with Transformer-based alternatives. Given Transformer's dominance in similar tasks, this comparison is critical.

2. Limited novelty in key components:Temporal Drift Compensation appears to be standard sliding window optimization integrated into the learning framework, lacking clear distinction from traditional loop closure methods

3. Incomplete ablation studies: Missing direct comparison replacing Mamba with Transformer/LSTM architectures

4. Unclear technical details: Deformable Mamba implementation is vaguely described - how exactly are deformable sampling and Mamba's SSM combined?

5. Missing comparisons with 2024-2025 state-of-the-art methods (e.g., MambaVO++ only appears in Table 2). The field evolves rapidly, and more current baselines would strengthen the evaluation.

**Questions:**

1.  Can you provide theoretical complexity analysis and direct experimental comparisons (accuracy and efficiency) between Mamba and Transformer-based implementations of your framework? What are the specific advantages of Mamba's selective SSM for odometry?

2. How does your drift compensation fundamentally differ from traditional sliding window bundle adjustment or pose graph optimization? Can you clarify the learning vs optimization components and what is end-to-end differentiable?

3. Table 5 shows 74ms inference, but how does memory consumption scale with history length T_h? What is the breakdown of computation time across modules (feature extraction, MMG, drift compensation)? How does performance scale to longer sequences?

4. Can you provide more details on how deformable attention sampling is integrated with Mamba's state space model? Are the sampling offsets learned jointly with the SSM parameters? Algorithm pseudocode would be very helpful here.

5. Temporal Drift Compensation appears to be standard sliding window optimization integrated into the learning framework, lacking clear distinction from traditional loop closure methods ？

---

### Official Review · Reviewer_fqw9 · 2025-11-02

**Soundness:** 3
**Presentation:** 3
**Contribution:** 2
**Rating:** 4
**Confidence:** 4

**Summary:**

In this paper, the authors propose a new approach (ST-VLO) based on Mamba for visual-lidar odometry with spatial-temporal correlation.

More specifically, ST-VLO consists a feature deformation module to sample visual features with reprojected point clouds. The sampled visual feature and point cloud features are fused for ego motion feature learning. The ego motion feature are also accumulated for long-term perception. Additionally, to reduce the error accumulation, the residual pose error is computed from two frames with predicted pose and then supervised with groundtruth. Finally, a keypoint-aware auxiliary loss is adopted to mitigate the influence of dynamic objects by focusing on top-k pose queries.

Experiments on public kitti and Argoverse datasets demonstrate the proposed method gives higher accuracy than previous approaches and is also faster than DVLO etc.

Ablation studies prove the effectiveness of the introduced components including the unification of lidar and visual data, motion compensation, and auxiliary loss.

**Strengths:**

1.	The alignment between lidar and visual inputs. Although Eq (1) is called deformable mamba in the paper, it is essentially an alignment between lidar and visual features. Obviously, it is very useful for lidar-visual odometry especially when the calibration parameters of lidar and camera are provided.

2.  The temporal drift compensation sounds interesting. The accumulated errors are computed from the warped point cloud with predicted pose and real point cloud. This allows the model to introduce geometric constraints as opposed to only pose constraints.

3.	The keypoint-aware auxiliary loss  is useful. Top-k queries with the smallest pose errors are used for final pose estimation enable the model itself to choose which keypoints are best for pose estimation, and thus could mitigate the influence of dynamic objects.

4.	Experiments on two public datasets are good and ablation studies look convincing.

5.	Comparison with previous approaches on running supports the claim of higher efficiency in the paper.

**Weaknesses:**

After reading this paper, I find it shares some similarities with previous works, but these similarities are not clearly described.

1.	The deformable mamba is essentially a lidar-visual alignment and is very straightforward. It can hardly be taken as a novelty. The temporal mamba which accumulates historical information to reduce error accumulation is also widely used in many previous VO frameworks using ConvGRU for sequence modeling [R1, R2]. Even the memory concept also sounds similar. However, these similarities are not discussed in the paper.

2.	The temporal drift compensation is another contribution of this work. However, what is the essentially difference with the pose refinement module in (Wang et al., 2021a)?


3.	In the paper, the authors use only one figure to support the claim that keypoint-aware auxiliary loss allows the model to focus mainly on the static areas and mitigate the influence of dynamic objects. This might not be enough to prove this claim. It would be better to give a quantitative results on the distribution of keypoints used for pose estimation.


[R1] Beyond tracking: Selecting memory and refining poses for deep visual odometry, Xue et al., CVPR 2019.

[R2] Deep visual odometry with adaptive memory, xue et al., TPAMI 2020.

**Questions:**

1.	Eq (2) and (3) allow the model to use accumulated history information to improve the current pose. However, newly gathered information (e.g. current frame) can not be used to improve previous poses. How can we do that?

2.	As mentioned in the weaknesses section, I would like to see more descriptions about the similarities of key components of this work and some previous works.

3.	The detailed analysis of keypoint-ware auxiliary loss and the proof would be more convincing.

---

### Note · Authors · 2025-11-12

I have read and agree with the venue's withdrawal policy on behalf of myself and my co-authors.

---

### Note · Authors · 2025-11-12

I have read and agree with the venue's withdrawal policy on behalf of myself and my co-authors.